# Effect of Plasma Activated Water Foliar Application on Selected Growth Parameters of Maize (*Zea mays* L.)

**Petr Škarpa [1],\*** , **Daniel Klofáč [1]**, **František Krčma [2]** , **Jana Šimečková [1]** and **Zdenka Kozáková [2]**

[1] Faculty of AgriScience, Mendel University in Brno, Zemědělská 1, 61300 Brno, Czech Republic; daniel.klofac@mendelu.cz (D.K.); jana.simeckova.uapmv@mendelu.cz (J.Š.)

[2] Faculty of Chemistry, Brno University of Technology, Purkyňova 118, 61200 Brno, Czech Republic; krcma@fch.vut.cz (F.K.); kozakova@fch.vutbr.cz (Z.K.)

\* Correspondence: petr.skarpa@mendelu.cz

**Abstract:** Utilization of plasma activated water (PAW) for plant growing is mainly connected with the treatment of seeds and subsequent stimulation of their germination. A potential of PAW is its relatively simple and low-cost preparation that calls for studying its wider application in plant production. For this purpose, a pot experiment was realized in order to prove effects of the foliar PAW application on maize growth. The stepped PAW foliar application, carried out in 7-day intervals, led to provable decrease of chlorophyll contents in leaves compared to the distilled water application. The PAW application significantly increased root electrical capacitance, but it had no provable effect on weight of the aboveground biomass. Chlorophyll fluorescence parameters expressing the $CO_2$ assimilation rate and variable fluorescence of dark-adapted leaves were provably decreased by PAW, but quantum yield of photosystem II electron transport was not influenced. A provably higher amount of nitrogen was detected in dry matter of plants treated by PAW, but contents of other macro- and micro-nutrients in the aboveground biomass of maize were not affected. Results of this pilot verification of the PAW application have shown a potential for plant growth optimization and possibility for its further utilization, especially in combination with liquid fertilizers.

**Keywords:** chlorophyll content; fluorescence parameters; root electrical capacitance; plant weight; macro- and micro-nutrient content

## 1. Introduction

Due to the rising population on the Earth, the question of food demand provision is becoming more and more important [1]. Besides conventional means of farming including application of pesticides and fertilizers, new ways for plant growing with as low impact on environment and non-target organisms as possible can be utilized. One of the new possibilities is the application of plasma. The first attempt with plasma is assigned to Henry Cavendish thanks to his work "Experiments on air" (1785) in which he tried to produce nitric acid by an electric spark in atmospheric air [2]. Nowadays, plasma has been utilized in the industry for tens of years [3], but the contemporary interest is focused on plasma applications in biomedicine and interdisciplinary fields connecting physics and biology [4–7].

This interest of the non-thermal plasma application (also called "cold" plasma due to its real neutral gas temperature) is still increasing and simultaneously, the number of experiments with plasma utilization for the treatment of plant materials is also increasing [8–15]. Therefore, a new research field called "Plasma agriculture" is arising [15,16]. Plasma potential for its use in agriculture is extensive. There are a lot of possibilities for its utilization, e.g., plasma treatment of seeds causing their disinfection [10,15,17] or stimulation of their germination capacity [10]. Using plasma, nitrogen from air can be captured, held and incorporated into water. This process leads to the formation of nitrogen

compounds capable to sustain plants [8,18,19] and reactive compounds of oxygen utilized for the stress decrease of pathogens in the soil [20]. Plasma can be also applied for disinfection of food packaging [21].

Plasma treatment can be realized by three ways. The first way is a direct irradiation by plasma, which provides the highest production of reactive species of oxygen (ROS) and nitrogen (RNS), ions, electrons and photons. The second way is based on a distant plasma interaction that also provides high production of ROS, RNS, ions and photons but with a lesser damage of the treated material. The third possibility is the application of plasma activated water (PAW). It provides a relatively low concentration of ROS and RNS with short lifetimes, but it produces a higher amount of N and O species with longer lifetimes whose application is accompanied by a low risk of the plant damage [9]. Plasma can interact with water directly (inside the bulk of water) or distantly (above the water surface) and water after the treatment is called the plasma activated water (PAW) [22,23].

Up until now, a lot of experiments were realized with the application of PAW on plant seeds in order to observe velocity of their germination capacity, sprouting and initial growth. In all cases, positive effects of PAW were confirmed [11–15].

One of important strategies of plant nutrition is based on the foliar application of nutrients/fertilizers. These compounds are commonly applied in water solution. A subject of this study is a pilot confirmation of the foliar PAW application on the plant growth. Due to its composition, PAW, as a relatively low cost and available water modification, is convenient for nitrogen donation to the plant growth. Moreover, contents of hydrogen peroxide and peroxonitrile acid determine its application as a prevention of fungal and bacterial diseases that increases efficiency of this application. Therefore, the PAW application seems to be a convenient arrangement combining effects of nutrition and plant protection by an environmentally friendly form utilized in ecological agriculture.

Based on the chemical composition of the plasma activated water, it can be assumed that its foliar application would influence a plant growth of maize, increase chlorophyll content and related parameters of photosynthesis, and increase the aboveground organs production of plants as well as subsequent root system capacitance, and influence content of nutrients in plants.

However, no experiments with a foliar application of PAW on plants and no observation of its effect on plant growth have yet been realized. The main aim of the presented study is a pilot verification of the effect of foliar applied PAW on selected growth parameters of maize plants. The above aim was fulfilled using a pot experiment established under controlled conditions in a growth box.

## 2. Materials and Methods

### 2.1. Plant Materials, Cultivation and Growth Conditions

An influence of the PAW application on growing parameters of maize was studied at conditions of a vegetative pot experiment established in the Biotechnological house at the Mendel University in Brno. The maize (*Zea mays* L.), cultivar SY ORPHEUS (Syngenta Czech s.r.o., Prague, Czech Republic) was chosen for this study. The pot experiment under the controlled day length, temperature and humidity (Figure 1) was carried out in the growth box (PlantMaster, CLF Plant Climatics GmbH, Wertingen, Germany). Plants were grown in 1.2 L plastic pots with 1000 g of arable soil. Chemical analysis of this soil is shown in Table 1.

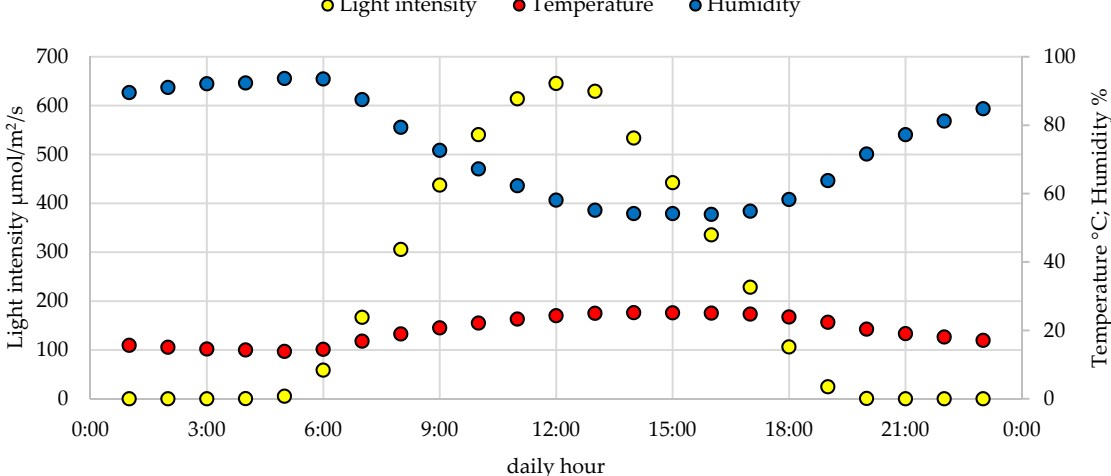

**Figure 1.** Settings of daily weather conditions of the pot experiment.

**Table 1.** Chemical composition of soil used in this study.

| Soil Parameter | Value |
|---|---|
| pH ($CaCl_2$) | 6.09 |
| Cox | 0.80% |
| Clay | 20% |
| Silt | 27% |
| Sand | 53% |
| Cation Exchange Capacity | 164 mmol/kg |
| N total | 0.19% |
| N-$NH_4^+$ ($K_2SO_4$) | 1.48 mg/kg |
| N-$NO_3^-$ ($K_2SO_4$) | 17.2 mg/kg |
| P (Mehlich 3) | 36.4 mg/kg |
| K (Mehlich 3) | 400 mg/kg |
| Ca (Mehlich 3) | 2720 mg/kg |
| Mg (Mehlich 3) | 214 mg/kg |

Four seeds of maize per pot were sown on 6 January 2020. In each case, plants were thinned to two plants per pot two weeks after the sowing. The first foliar application of distilled water (DW) or plasma activated water (PAW) was carried out on the growth stages of three leaves (T0). The second application was performed seven days after the first one (T1), the third application (T2) followed one week later. The following six treatments with three doses/levels of water (1, 2 and 3) were included in the experiment: DW1—one DW foliar application (T0); DW2—two DW foliar applications (T0 and T2); DW3—three DW foliar applications (T0, T1 and T2); PAW1—one PAW foliar application (T0); PAW2—two PAW foliar applications (T0 and T2); PAW3—three PAW foliar applications (T0, T1 and T2). The PAW and DW as a foliar spray at 3 mL per pot at each of three application terms were used. PAW and DW were evenly applied using a pressurized hand pump. Each treatment was performed in eight replications (pots).

### 2.2. PAW Preparation

Plasma activated water (PAW) was prepared in a dielectric barrier discharge (DBD) plasma system [24] that consisted of a Pyrex glass Petri dish (90 mm in diameter) with a graphite outer electrode and the other electrode made of a ceramic plate with an upper silver. Both electrodes were connected to the power supply (Lifetech s.r.o., Brno, Czech Republic) operating at frequency of 11 kHz and giving a sinusoidal peak-to-peak voltage of 16 kV. Total energy supplied from the electrical network was (36 ± 2) W. A quantity of 75 mL of distilled water was treated 8 times for 15 s. This water volume ensured a gaseous gap of 3.2 mm between the distilled water surface and the upper ceramic electrode.

Because of the limited volume of the plasma reactor, the prepared PAW was homogenized by mixing of all doses together to decrease the PAW quality dispersion. The PAW amount needed for each application was prepared immediately before its use at ambient laboratory conditions.

The chemical activity of PAW was evaluated by the determination of the produced amount of hydrogen peroxide, nitrites and nitrates. Concentration of hydrogen peroxide was determined calorimetrically using the titanium reagent [25]. Concentrations of nitrites and nitrates were measured using the commercial $NO_x$ test kits (TetraTest $NO_2^-$ and TetraTest $NO_3^-$) that are based on the Griess reaction [26]. The mean concentration of hydrogen peroxide in PAW was $(0.7 \pm 0.2)$ mg/L $((0.022 \pm 0.004)$ mmol/L), the concentration of nitrites was $(1.071 \pm 0.005)$ mg/L, and nitrates $(24.7 \pm 2.3)$ mg/L. Concentration of hydrogen peroxide, nitrites as well as nitrates in DW (i.e., without the plasma activation) was under the detection limit.

### 2.3. Measurement of Selected Growth Parameters

During the maize vegetation period, selected growth parameters were evaluated within the pot experiment. Namely, chlorophyll content (N-tester value), fluorescence parameters (quantum yield of photosystem II, variable fluorescence of dark-adapted leaves, a chlorophyll fluorescence decrease ratio), root electrical capacitance, weight of dry matter and nutrient content in aboveground biomass of plant. Times of measurements were different according to each parameter. They were set in a 7-day interval coincidently with the second and third DW and PAW application (T1, T2) and two following terms (T3 and T4). The measurement of the studied parameters always came before the water application.

#### 2.3.1. Chlorophyll Content (N-Tester Value)

The contents of leaf chlorophyll were measured with a hand-held chlorophyll meter. Relative indices for the chlorophyll contents of leaves (N-tester values) were obtained with the N-Tester chlorophyll meter from Yara (Yara International ASA, Oslo, Norway). The measurement is based on leaf transmittance of the wavelengths 650 nm (red) and 940 nm (IR) [27]. Four plants (two leaves from the lowest part) were assessed in each treatment. The measurement was performed on the leaves of the whole plant, the value of chlorophyll content of each plant is the mean of 30 measurements.

#### 2.3.2. Chlorophyll Fluorescence Parameters

Photochemical quenching parameters that measure the efficiency of the photosystem II (PSII) photochemistry were determined (terms T1–T4). These parameters measure the proportion of the light absorbed by chlorophyll associated with PSII. Two leaves on each experimental plant (from the lowest part of the plant) were labeled for the measurement. Determination of chlorophyll fluorescence was carried out on the same labeled leaves by the fluorometer PAR-FluorPen FP 110-LM/S (Photon Systems Instruments, Drásov, Czech Republic) and the software FluorPen 1.1 was used for the analysis of the measured data. The plants were adapted for 1 h in the dark before the measurement of chlorophyll fluorescence. Characteristics of the measurement protocol of the chlorophyll fluorescence are presented in Table 2.

**Table 2.** Measurement protocol of the chlorophyll fluorescence.

| Chlorophyll Fluorescence Parameters | Pulse Type | Light Inetnsity ($\mu$mol/m$^2$/s) | Phase | Duration (s) | 1st Pulse (s) | Pulse Interval (s) |
|---|---|---|---|---|---|---|
| $\Phi_{PSII}$, $F_v$ | Saturation | 2400 | - | 1 pulse | | |
| $R_{Fd}$ | Flash | 900 | L | 60 | 0.2 | 1 |
| | | | DR | 88 | 1 | 1 |
| | Saturation | 2400 | L | 60 | 7 | 12 |
| | | | DR | 88 | 11 | 26 |
| | Actinic | 300 | L | 60 | - | - |

$\lambda = 454$ nm, L—light, DR—Dark recovery.

The following chlorophyll fluorescence parameters were measured: quantum yield of the PSII ($\Phi_{PSII}$) as the equivalent to $F_v/F_m$ in the dark-adapted leaves that express efficiency of the photosystem II [28]; variable fluorescence of the dark-adapted leaves ($F_v$) calculated from the relation $F_m$-$F_0$ ($F_0(m)$ means minimal(maximal) fluorescence from the dark-adapted leaves) [29]; and the chlorophyll fluorescence decrease ratio ($R_{Fd}$) as the ratio $F_d/F_s$, which is directly proportional to the net $CO_2$ assimilation rate of leaves and expresses efficiency of the photosystem ($F_d$ means fluorescence decrease from $F_m$ to $F_s$; $F_s$ means steady state chlorophyll fluorescence) [30].

### 2.3.3. Root Electrical Capacitance

The root electrical capacitance measurements were carried out using a simple portable hand instrument VOLTCRAFT LCR 4080 (Conrad Electronic GmbH, Wels, Austria). The electrical capacitance was measured in nanofarads (nF) using a frequency signal of 1 kHz. The ground electrode (20 cm long stainless-steel) was inserted vertically to the 10 cm depth in the soil, 4 cm far from the stem of the maize. The plant electrode was Al-clamp fixed 2 cm above the soil surface. This instrument was used for the measuring of the root capacitances at two growth stages of plants (T2 and T4) in the pots after the irrigation (under condition of 100% of the full water capacity).

### 2.3.4. Plant Analysis

The weight of the dry matter (DM) and the content of macronutrients (N, P, K, Ca, Mg) and micro-nutrients (Zn, Mn, Fe) were determined in the plant aboveground biomass in the last term of the plant sampling (T4). The samples of plant mass were dried at temperature of 60 °C, then crushed in a grinder, and homogenized. After the $HNO_3/H_2O_2$ microwave digestion in ETHOS 1 (Milestone Srl, Sorisole, Italy), the content of nutrients was determined. The content of P was determined colorimetrically using the Unicam 8625 UV/Vis spectrometer (Pye Unicam Ltd, Cambridge, UK). The contents of K, Ca, Mg, Cu, Fe, Zn and Mn were determined by the atomic absorption spectrometry (AAS) with the ContrAA 700 instrument (Analytik Jena AG, Jena, Germany). The amount of N was determined using the Kjeldahl method in the Vapodest 50s analyser (C. Gerhardt GmbH & Co. KG, Königswinter, Germany).

### *2.4. Data Analysis*

Statistical evaluation of the monitored parameters was performed by the Statistica 12 CZ program [31]. Two-way ANOVA analysis of variance and follow-up tests according to Fisher (LSD test) at the 95% ($p < 0.05$) level of significance were used and the results were expressed as the mean ± standard error (SE).

## 3. Results and Discussion

### *3.1. The Effect of PAW on Chlorophyll Contents (N-Tester Value)*

The mean content of chlorophyll (N-tester value) in the maize leaves treated by PAW was not significantly ($p \leq 0.05$) different from the values measured after the foliar treatment with DW. The N-tester values reached the level of 97.7% for the treatment variants with the foliar DW application. The mean N-tester values of individual experimental variants are presented in Table 3. Comparing both tested water types (DW and PAW), provable differences were determined for N-tester values between the doses/intensity of water application ($p \leq 0.05$). Whereas the stepped DW application decreased the N-tester values but with no significant trend, the PAW application has enhanced the N-tester values significantly. By the comparison of variant PAW3 to PAW1, the N-tester value was enhanced significantly ($p \leq 0.05$) by 6.5%. This effect is probably related to the presence of nitrogen (in a form of nitrates and nitrites) in the PAW that has led to the increased chlorophyll production [32,33].

**Table 3.** The effect of plasma activated water (PAW) application on chlorophyll contents (N-tester value).

| Treatment | | N-Tester Value |
|---|---|---|
| Mean value of water type | DW | 420 [a] ± 11 |
| | PAW | 414 [a] ± 10 |
| Mean value of treatment variant | DW1 | 418 [a] ± 19 |
| | DW2 | 429 [ab] ± 15 |
| | DW3 | 413 [a] ± 20 |
| | PAW1 | 404 [a] ± 19 |
| | PAW2 | 408 [a] ± 18 |
| | PAW3 | 431 [b] ± 14 |

The mean values sharing the same superscript are not significantly different from each other ($p \leq 0.05$) according to the LSD test. The values represent the mean (n = 60) ± standard error (SE).

Time evaluation of N-tester values for both water types was different. Whereas the N-tester values for plants treated by DW and PAW were not remarkably varied in the first term of the measurement (T1), they were significantly ($p \leq 0.05$) different in the following terms. A decrease of the N-tester values in time was higher for variants with the enhanced dose/intensity of DW application. Chlorophyll contents of the variant DW3 expressed by the N-tester value was decreased from the original 483 [c] ± 6 to 316 [a] ± 8 within three weeks, i.e., by 34.6% (Figure 2). By the PAW application of the same doses/intensity, the N-tester values were also decreased in time (from 471 [c] ± 6 to 355 [b] ± 3) but it represented the decrease of 24.7%, only. This phenomenon is probably related to the previously mentioned contents of nitrates in the PAW and its influence on the chlorophyll formation. This fact is amplified by the N-tester values determined for PAW variants in the last term (T4), which are enhanced by the stepped PAW application. A fertilization effect of PAW applied into the soil, that is stimulated by the presence of nitrates and which positively influences the plant growth, is mentioned in other studies [8,34,35]. Adhikari et al. [36] also noticed a change of chlorophyll contents after the PAW application in the dependence on the PAW preparation.

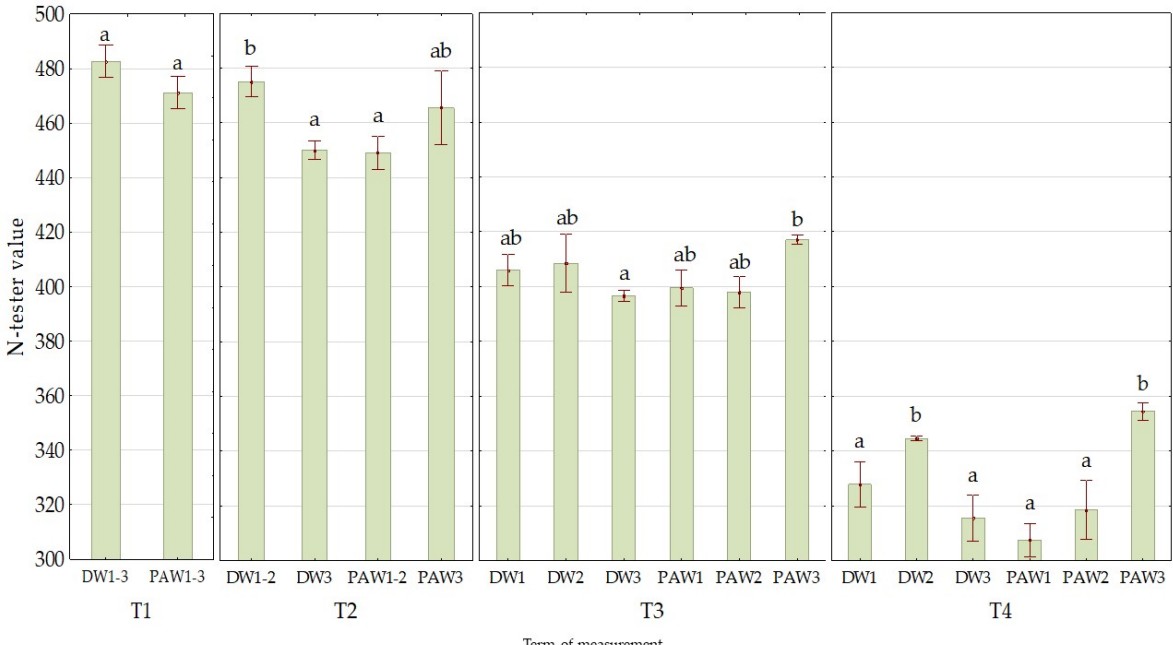

**Figure 2.** The effect of PAW application on chlorophyll contents (N tester value) over time. The mean values (n = 60) sharing the same lowercase letter are not significantly different from each other ($p \leq 0.05$) according to the LSD test (each term was evaluated separately). Intervals present the mean standard error (SE).

### 3.2. The Effect of PAW on Chlorophyll Fluorescence Parameters

The quantum yield of the electron transport of the photosystem II ($\Phi_{PSII}$), that indicates an actual capacity of the photosystem II (PSII) for photochemical processes by availability of reaction centers of the photosystem II, was not significantly ($p \leq 0.05$) influenced by the PAW application. The stepped application of DW did not significantly ($p \leq 0.05$) influence the quantum yield of PSII. On the other hand, its value was provably increased by the dose/intensity of the PAW2 variant (Table 4). However, further increase of the PAW doses/intensity (PAW3) did not lead to any significant change of the quantum yield values. Fryer et al. [37] mentioned a strong linear relationship between $\Phi_{PSII}$ and the efficiency of carbon fixation, especially at the laboratory conditions. Kučerová et al. [38] reported on higher content of photosynthetic pigments of wheat plants, which were irrigated with the PAW. Watering by PAW increases the concentration of photosynthetic pigments and simultaneously has no or negative impact on net photosynthesis of barley and maize [39].

**Table 4.** Influence of PAW application on chlorophyll fluorescence parameters.

| Treatment | | $\Phi_{PSII}$ | $F_v$ | $R_{Fd}$ |
|---|---|---|---|---|
| Mean value of water type | DW | 0.759 [a] ± 0.002 | 1022 [b] ± 8 | 0.51 [a] ± 0.04 |
| | PAW | 0.755 [a] ± 0.003 | 973 [a] ± 7 | 0.57 [b] ± 0.05 |
| Mean value of treatment variant | DW1 | 0.759 [ab] ± 0.003 | 1022 [c] ± 16 | 0.44 [a] ± 0.07 |
| | DW2 | 0.758 [ab] ± 0.003 | 1033 [c] ± 12 | 0.53 [bc] ± 0.08 |
| | DW3 | 0.758 [ab] ± 0.003 | 1009 [bc] ± 12 | 0.54 [bc] ± 0.10 |
| | PAW1 | 0.749 [a] ± 0.006 | 980 [ab] ± 13 | 0.61 [c] ± 0.08 |
| | PAW2 | 0.760 [b] ± 0.002 | 959 [a] ± 11 | 0.60 [c] ± 0.08 |
| | PAW3 | 0.756 [ab] ± 0.004 | 978 [a] ± 11 | 0.49 [ab] ± 0.09 |

The mean values sharing the same superscript are not significantly different from each other ($p \leq 0.05$) according to the LSD test (each column was evaluated separately). The values represent the mean (n = 8) ± standard error (SE).

Variable fluorescence of the dark-adapted leaves ($F_v$), which expresses ability of the photosystem II to absorb radiation, was significantly ($p \leq 0.05$) decreased by the PAW application. The mean value of all tested variants (PAW1−3) reached 95% of the value determined after the DW application. The ability of plants to absorb radiation was reduced by the PAW application, as it is demonstrated in Table 4. The significantly ($p \leq 0.05$) highest values of variable fluorescence were achieved by the application of DW (DW1 and DW2).

The application of the tested water types had a different influence on the fluorescence decrease ratio ($R_{Fd}$), which is measured at the saturation irradiance and is directly proportional to the net $CO_2$ assimilation rate. Whereas the stepped doses/intensity of the DW application led to a significant ($p \leq 0.05$) increase of the $R_{Fd}$ value that has reached for the variant DW3 the value of 121.7% compared to DW1, the stepped PAW application had decreased this parameter. The highest dose/intensity of the PAW application (PAW3) led to the provable reduction of the $R_{Fd}$ value with the decrease of 19.4% comparing to the PAW1 variant.

Although the mean values of $\Phi_{PSII}$ were provably ($p \leq 0.05$) enhanced by the double dose/intensity application of plasma activated water (PAW2), from the value evaluation of the photosystem II capacity, it is evident for the photochemical processes that the application of DW and PAW had no provable ($p \leq 0.05$) influence on the availability of reaction centers of the photosystem II in individual observation terms, as it is demonstrated in Figure 3.

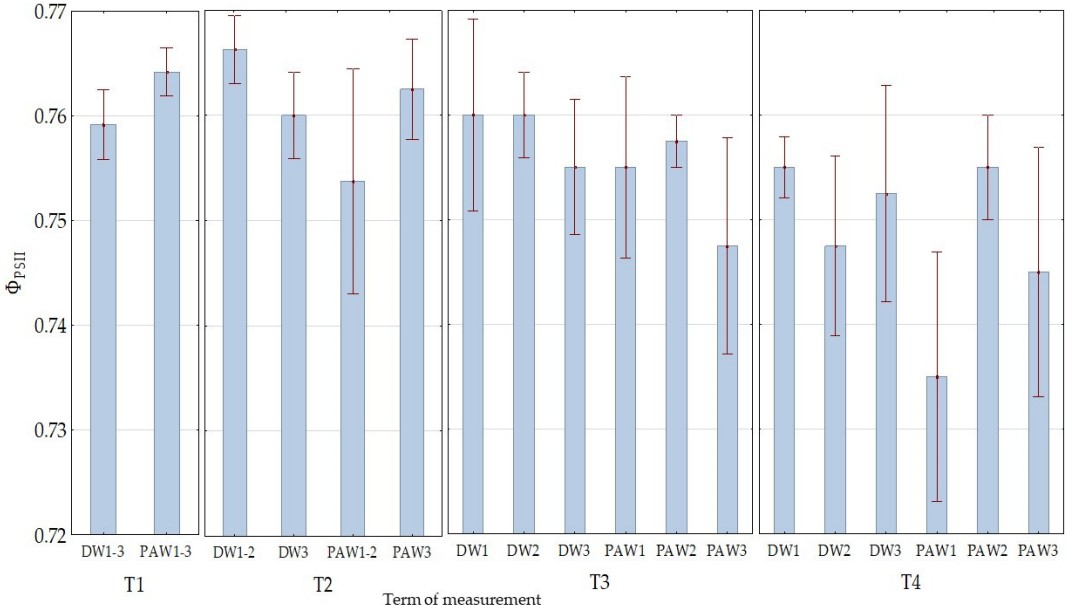

**Figure 3.** Effect of PAW on quantum yield of PSII ($\Phi_{PSII}$) over time. Lack of lowercase lettering indicates no significant differences between the mean values (n = 8). Intervals present the mean standard error (SE).

Variable fluorescence, which is determined by the redox state of the primary quinone electron acceptors of the PSII [40], was significantly ($p \le 0.05$) the highest for variants DW1 and DW2 in the most of the observation terms. The values of variable fluorescence maximal yield in the dark-adapted leaves measured in all terms did not differ between the stepped PAW doses (Figure 4), but their value was provably ($p \le 0.05$) lower in the last two terms (T3 and T4) compared to the plants treated by DW. This decrease can be caused by the presence of hydrogen peroxide ($H_2O_2$) in the PAW which decreases the maximal photochemical efficiency of the PSII even at its low concentration as well as the maximal quantum yield of primary photochemistry and other photosynthetic parameters [41].

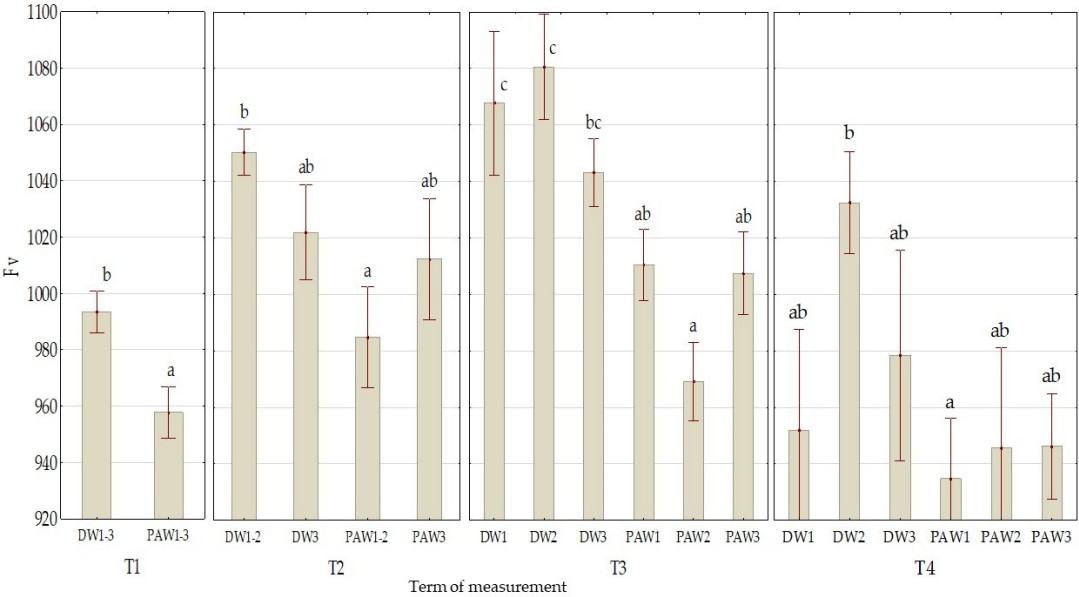

**Figure 4.** Change of variable fluorescence ($F_v$) value after the PAW application over time. The mean values (n = 8) sharing the same lowercase letter are not significantly different from each other ($p \le 0.05$) according to the LSD test (each term was evaluated separately). Intervals present the mean standard error (SE).

The chlorophyll fluorescence decrease ratio ($R_{Fd}$) correlates with the potential $CO_2$ fixation rate of leaves as was shown for several plants as well as sun and shade leaves [30]. Values of $R_{Fd}$ below 1.0 suggest disruption in $CO_2$ assimilation [42]. The values of the fluorescence decrease ratio were substantially different in the observation terms (Figure 5). In the early terms (T1 and T2), the PAW application had a positive effect on the $CO_2$ fixation evaluated as the $R_{Fd}$ value. Especially in the term T2, the $R_{Fd}$ value was significantly ($p \leq 0.05$) enhanced when the PAW variant was used instead of DW. Specifically, the $R_{Fd}$ value was increased from 0.27 [a] ± 0.01 to 0.50 [c] ± 0.05 by the PAW application of 1–2 doses whereas it was increased from 0.31 [ab] ± 0.07 to only 0.40 [bc] ± 0.03 by the higher PAW application of 3 doses. However, even in this term it is evident that the $R_{Fd}$ value is decreased with the rising doses/intensity of the PAW application (PAW3) and this decrease is further deepened in the following terms. Based on these results it can be assumed that the intensive PAW application on the maize plants leads to the damage of the photosystem apparatus and to the subsequent reduction of the $CO_2$ fixation by the plants. This presented fact correlates with the weight of the dry matter determined in the last experimental term (T4). Contrary to this result, the PAW application in the low or middle intensity (PAW1 and PAW2, respectively) leads to the enhancement of the $R_{Fd}$ value if we omit the data obtained in the last but one term (T3). These values from the early terms were significantly ($p \leq 0.05$) the highest at the end of the experiment. An opposite trend was observed by the application of DW. Comparing to the one dose DW application (DW1), significantly ($p \leq 0.05$) higher $R_{Fd}$ values were achieved with the stepped DW application (DW2 and DW3) at the end of the experiment (T4).

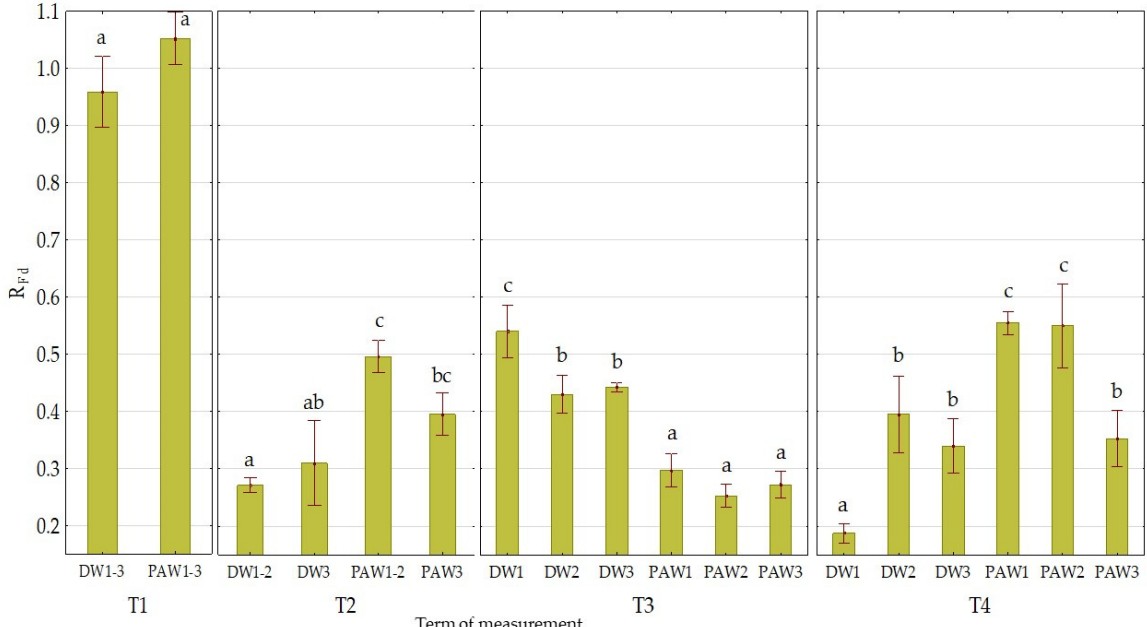

**Figure 5.** Change of Fluorescence decrease ratio ($R_{Fd}$) after PAW application over time. The mean values (n = 8) sharing the same lowercase letter are not significantly different from each other ($p \leq 0.05$) according to the LSD test (each term was evaluated separately). Intervals present the mean standard error (SE).

*3.3. The Effect of PAW on Root Electrical Capacitance ($C_R$)*

Evaluation of the DW and PAW application effect on the root electrical capacitance of maize was realized in the term T2 and at the end of the experiment (T4). A close relationship between the value of the electrical capacitance of the root and its length and weight has been proven and published many times e.g., [43,44]. The mean $C_R$ values for both tested types of water and their doses/intensity of application are presented in Table 5. The foliar application of the PAW significantly ($p \leq 0.05$) increased the $C_R$ value by 14.6% compared to the DW treatment. An enhanced root production

during the early developmental stages after the PAW application was observed at *Vigna radiata* [45] or *Nicotiana tabacum* [46]. A significant increase of the root weight of *Vigna mungo* whose seeds were treated by the PAW was reported in [47]. The enhancement of the mean values of the root capacitance was observed after the application of both types of water. Comparing the first and third level of the doses/intensity of application, the relative increase reached 6.9% by the DW application and 26.8% by the PAW application, respectively (Table 5).

**Table 5.** Influence of PAW application on root electrical capacitance ($C_R$) in nano-farads (nF).

| Treatment | | T2 | T4 | Mean $C_R$ |
|---|---|---|---|---|
| Mean value of water type | DW | - | - | 0.192 [a] ± 0.003 |
| | PAW | - | - | 0.220 [b] ± 0.006 |
| Mean value of treatment variant | DW1 | 0.20 [a] ± 0.01 | 0.18 [a] ± 0.01 | 0.188 [a] ± 0.004 |
| | DW2 | | 0.19 [ab] ± 0.01 | 0.188 [a] ± 0.005 |
| | DW3 | 0.20 [a] ± 0.00 | 0.20 [ab] ± 0.01 | 0.201 [ab] ± 0.005 |
| | PAW1 | 0.19 [a] ± 0.01 | 0.21 [bc] ± 0.01 | 0.198 [ab] ± 0.006 |
| | PAW2 | 0.23 [b] ± 0.01 | 0.23 [c] ± 0.01 | 0.212 [b] ± 0.009 |
| | PAW3 | | 0.27 [d] ± 0.02 | 0.251 [c] ± 0.011 |

The mean values sharing the same superscript are not significantly different from each other ($p \leq 0.05$) according to the LSD test (each column was evaluated separately). The values represent the mean (n = 8) ± standard error (SE).

### 3.4. The Effect of PAW on Dry Matter Weight

The dry weight of the aboveground organs of maize, which was determined only in the last experimental term (T4), was not influenced by the foliar PAW application ($p \leq 0.05$). Nevertheless, from the mean values it is evident that the weight of the dry matter of the aboveground mass was decreased by the increasing doses/intensity of application. The lowest value among all variants was achieved with the variant PAW3 (Table 6). The available results indicate that the PAW used for watering of germinated plants of wheat effectively stimulate seedlings growth and positively affect their metabolism in the soil with a low nutrient content [38]. An opposite trend was observed by the DW application when the weight of the dry matter was increased. The results of Fan et al. [45] indicates that time of the PAW preparation determines its effect on the plant growth. While the application of the PAW exposed to the plasma for 15 min stimulated the plant weight and stem length of *Vigna radiata* seedlings, the PAW exposed to the plasma for a long time decreased their weight. Sajib et al. [47] also reports the effect of the PAW treatment of the *Vigna mungo* seeds on the plant growth and weight of the aboveground mass in the dependence on the plasma treatment durations. Based on the available results dealing with the PAW effect on the plant growth we can suppose that the higher PAW dose/intensity of application (PAW2 and PAW3) could be the cause of the decreased weight of the aboveground plant mass even by its foliar application.

**Table 6.** Influence of PAW application on dry matter (DM) weight (g/plant).

| Treatment | | DM (g/plant) |
|---|---|---|
| Mean value of water type | DW | 0.546 ± 0.007 |
| | PAW | 0.529 ± 0.010 |
| Mean value of treatment variant | DW1 | 0.534 ± 0.010 |
| | DW2 | 0.544 ± 0.027 |
| | DW3 | 0.559 ± 0.016 |
| | PAW1 | 0.548 ± 0.020 |
| | PAW2 | 0.521 ± 0.015 |
| | PAW3 | 0.519 ± 0.015 |

The values represent the mean (n = 8) ± standard error (SE). Lack of lettering indicates no significant differences between the mean values.

### 3.5. The Effect of PAW on Nutrients Content in Plant Tissue

Plasma activated water contains ions $NO_3^-$ and $NO_2^-$. Therefore, its stepped application significantly ($p \leq 0.05$) increased content of nitrogen in the plant tissue between the variant PAW1 by (1.43 [a] ± 0.05)% and PAW3 by (1.62 [a] ± 0.07)%. Nitrogen increase in the dry matter of the aboveground biomass was 13.3% (Figure 6a). Although the nitrate ion provided by the foliar application is a worse form for the plant intake [48] and compared to another forms, its absorption through the cuticle is worse than absorption of amides (urea) or ammonia [49], the foliar fertilization by water solutions has a promising potential to increase content of nitrogen in the plant. Peuke et al. [50] reports that the foliar applied inorganic N, both in the form of nitrate and ammonium, was entirely assimilated in the shoots of *Ricinus communis*. Our results with the enhanced nitrogen content in the maize plant were probably influenced by the lower weight of the dry matter in the variant PAW3, which decreased the dilution effect of the utilized nitrogen [51]. Contents of the rest macro-nutrients varied within the ranges: (0.15–0.17)% of P, (2.23–2.45)% of K, (1.09–1.22)% of Ca and (0.36–0.40)% of Mg. By the PAW application, only the content of P was significantly ($p \leq 0.05$) enhanced compared to the plants treated by DW (specifically DW2). Nevertheless, the mean content of phosphorus in plants treated by the PAW (0.17 [a] ± 0.00)% was almost the same as the mean content in plants treated by DW (0.16 [a] ± 0.00)%.

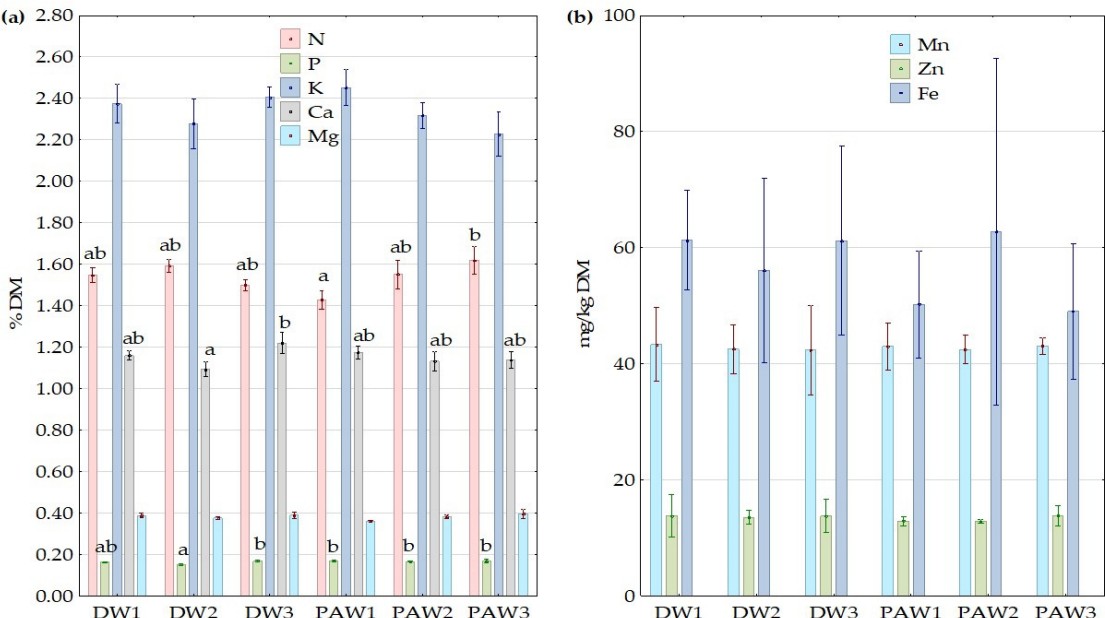

**Figure 6.** Content of (**a**) macro- and (**b**) micro-nutrients in above ground-biomass of plant. The mean values (n = 8) sharing the same letter are not significantly different from each other ($p \leq 0.05$) according to LSD test (each nutrient was evaluated separately). Lack of lettering indicates no significant differences between the mean values. Intervals present the mean standard error (SE).

Among the evaluated micro-nutrients in the plant, zinc was detected for its importance in the maize nutrition and its positive effect on the seed production is described by many authors e.g., [52,53]. Besides other functions, zinc represents an important cofactor for the formation of enzymes and proteins and significantly influences nitrogen metabolisms in plants [54], likewise manganese [55] or iron [56]. Contents of studied micro-nutrients was not significantly ($p \leq 0.05$) influenced by the water application (DW and PAW), as it is demonstrated in Figure 6b.

The supposed effect of the PAW application as the source of the foliar applied nitrogen, by which metabolisms of the plant influences the intake and utilization (content) of other nutrients, was not confirmed by the experiment.

## 4. Conclusions

Most of studies dealing with the PAW utilization in the plant production have tested its influence during the stage of germination or sprouting, whereas realization was provided especially by the seed dipping or soil watering. There are almost no available data about the effect of its foliar application on the plant growth during the vegetation period. Due to the PAW composition, its application has a high potential in the plant nutrition. The periodical PAW application positively influenced the chlorophyll content in the maize leaves. Content of chlorophyll expressed by the N-tester value was decreased in time at the highest distilled water application by 34.6% and by 24.7% at the PAW application of the same doses. The foliar PAW application significantly increased the root electrical capacitance, contrary to the weight of the aboveground biomass and the contents of nutrients in its dry matter except nitrogen. The periodical PAW application increased its content in the dry matter of the aboveground organs by 13.3%. The fluorescence parameters, that evaluate the efficiency of the photosystem II, were not significantly influenced by the PAW application either. Further research is supposed to be focused on the verification of the foliar PAW application in the plant protection and alternatively, as a substitution of water conventionally used for nutrient provision by the foliar fertilization.

**Author Contributions:** Conceptualization P.Š., and D.K.; methodology P.Š., D.K., and F.K.; formal analysis P.Š., D.K., F.K., and Z.K.; investigation P.Š., D.K., and F.K.; writing—original draft preparation, review and editing P.Š., Z.K., D.K., F.K and J.Š. All authors have read and agreed to the published version of the manuscript.

**Funding:** This research was financially supported by the Internal Grant Agency of Faculty of AgriSciences Mendel University in Brno under the project No. AF-IGA2020-IP085.

**Acknowledgments:** This research was supported by the Internal Grant Agency of Faculty of AgriSciences Mendel University in Brno under the project No. AF-IGA2020-IP085.

**Conflicts of Interest:** The authors declare no conflict of interest.

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
