# Peer review of "Effect of Plasma Activated Water Foliar Application on Selected Growth Parameters of Maize (Zea mays L.)"

_water, doi:10.3390/w12123545_

Round 1

Reviewer 1 Report

The article presents the results of influencing of maize by plasma activated water - PAW. Manuscript deals with interesting topic and is well written. However, I have two major questions to the authors which should be cover up in the text.

  • There is a doubt if all the presented effects are caused by the PAW or by the H2O2, HNO3, and HNO2. The comparison of application of PAW and artificial PAW (the solution of H2O2, HNOx) should be added.
  • Reviewer did not apprehend why the electric capacity was measured.

After answer, I recommend the publication of presented manuscript.

Author Response

Dear Reviewer

Thank you for your very careful review of our paper, and for the comments, corrections and suggestions. Below are the specific answers to the comments:

Referee 1

The article presents the results of influencing of maize by plasma activated water - PAW. Manuscript deals with interesting topic and is well written. However, I have two major questions to the authors which should be cover up in the text.

  1. There is a doubt if all the presented effects are caused by the PAW or by the H2O2, HNO3, and HNO2. The comparison of application of PAW and artificial PAW (the solution of H2O2, HNOx) should be added.

This is prepared for the separate experiment. We plan to get application of all species mentioned above individually as well as in synergy in the same concentrations as were measured.

  1. Reviewer did not apprehend why the electric capacity was measured.

We thank the Reviewer for this comment.

The growth of the root system is not necessary connected only with a direct contact of the tested agent (substrate, soil, watering) with the particular organ, i.e. the root in this case. The growth can be also influenced by the application of the agent on leaves. Effects of the foliar application (phytohormons, humic matters, biostimulaters, etc.) on the root growth (its length, size, and structure), the shoot/root ratio and other indicators are mentioned by many available literal sources. By these reasons, the authors have focused on the determination of the root size by a non-destructive method (electric capacitance). An influence of the foliar PAW application on the growth of the aboveground mass can be related with the effect of the application on the formation of the root system.

Reviewer 2 Report

The concept of this research article is quite interesting. However, in my opinion, the results of this study are not suitable for publication in the current form and the manuscript needs major revision. Some additional specific comments about the content are listed below.

  1. Keywords should not overlap with the title.
  2. Introduction: During plasma treatment of the material rather protons than photons can be produced.
  3. L73 – should be: Zea mays
  4. There is no research hypothesis.
  5. M&M: What volume of water (DW or PAW) was used for spraying of the plants? What equipment was used for spraying?
  6. The concentration of hydrogen peroxide, nitrites, and nitrates should be also measured in DW to compare its properties with PAW. It cannot be assumed in advance that these substances are not present in the DW - it must be checked.
  7. The chlorophyll concentration cannot be measured by N-tester. The N-tester measures the nitrogen content, not the chlorophyll content. The manufacturer's website says: ‘In theory N-Tester could be used on any plant to measure the chlorophyll content of the leaf and as such could be a useful management tool, however, no other crop calibrations exist so these would need to be developed locally by the user and any recommendations would need to be confirmed by locally conducted trials.’ In scientific research, such equipment for determining the concentration of chlorophyll should not be used.
  8. I don't think the device used (VOLTCRAFT LCR 4080) is capable of measuring the electrical conductivity of the roots. Firstly, it is not dedicated to this type of measurement. Secondly, in other works using this device is an estimated root system size (https://doi.org/10.11118/actaun201866040871). At what level of soil moisture were these measurements carried out? The soil water content strongly influences the measurement results. Moreover, the measured values relate to the rhizosphere rather than the roots themselves.
  9. Statistical analysis: maybe two-way ANOVA should be performed for all parameters tested, as experiments were established with two types of water (DW or PAW) and three number of water applications as experimental factors. In the methods authors must specify what type of ANOVA was performed.
  10. L173-174 – the authors wrote: ‘While the N-tester value was decreased by the increasing intensity of the foliar DW application..’ – it is not true, statistical analysis does not support this statement.
  11. L203-204 – ‘Its mean value measured after the PAW application in all evaluated terms was almost identical with its value obtained with the DW application.’ – this sentence is redundant.
  12. L281-282 – the authors wrote: ‘The enhancement of the mean values of the root capacitance was observed after the application of both types of water’, but what could be the reason for this? maybe the water from the leaves was just dripping down to the soil substrate?
  13. L289- ‘The weight of the dry matter of the aboveground maize mass…’ – please, edit this part of the sentence, you can write it more simply.
  14. The authors wrote: ‘Nevertheless, from the mean values it is evident that the weight of the dry matter of the aboveground mass was decreased by the increasing doses/intensity of application.’; ‘An opposite trend was observed by the DW application when the weight of the dry matter was increased.’; ‘…higher PAW dose/intensity of application (PAW2 and PAW3) could be the cause of the decreased weight of the aboveground plant mass even by its foliar application.’; L318 ‘… lower weight of the dry matter in the variant PAW3….’- statistical analysis does not confirm these claims.
  15. Figures and tables: The letters and numbers in the figures are poorly visible. Their quality should be improved. Moreover, in my opinion, when there are no statistical differences between the means, it is better not to mark the means with letters, it creates unnecessary confusion. It is better to explain in the description that the lack of lettering indicates no significant differences.
  16. It is not known from how many data the means were calculated. The value of ‘n’ (e.g. n=30, n=4) should be provided in descriptions of tables and figures.
  17. Conclusions should not be a summary of the results. It should be emphasized what is the novelty of this research and what practical guidelines can be forwarded. This part is definitely too long.

Author Response

Dear Reviewer

Thank you for your very careful review of our paper, and for the comments, corrections and suggestions. Revisions to the paper have been made taking most of the above into account. We believe that this has significantly improved our manuscript. Below are the specific answers to the comments:

Referee 2

The concept of this research article is quite interesting. However, in my opinion, the results of this study are not suitable for publication in the current form and the manuscript needs major revision. Some additional specific comments about the content are listed below.

  1. Keywords should not overlap with the title.

Redundant keywords were deleted.

  1. Introduction: During plasma treatment of the material rather protons than photons can be produced.

The term “photons” is correct. Protons are included in the term “ions”. Photons are generated by plasma as well and they represent the radiation from plasma.

  1. L73 – should be: Zea mays

It was corrected in all text of article.

  1. There is no research hypothesis.

The research hypothesis was performed and added to the manuscript added to the manuscript.

“One of important strategies of the plant nutrition is based on the foliar application of nutrients/fertilizers. These compounds are commonly applied in water solution. A subject of this study is a pilot confirmation of the foliar PAW application on the plant growth. Due to its composition, PAW, as a relatively low cost and available water modification, is convenient for nitrogen donation to the plant growth. Moreover, contents of hydrogen peroxide and peroxonitrile acid determine its application as a prevention of fungal and bacterial diseases that increases efficiency of this application. Therefore, the PAW application seems to be a convenient arrangement combining effects of nutrition and plant protection by an environmentally friendly form utilised in ecological agriculture.”

  1. M&M: What volume of water (DW or PAW) was used for spraying of the plants? What equipment was used for spraying?

We thank the Reviewer for this observation; Done. In the revised version of the manuscript we have added the information’s about the volume of water (DW or PAW) was used for spraying and used equipment for spraying. 

“The PAW and DW as a foliar spray of 3 mL per pot at each of three application terms were used. PAW and DW were evenly applied using a pressurized hand pump.”

  1. The concentration of hydrogen peroxide, nitrites, and nitrates should be also measured in DW to compare its properties with PAW. It cannot be assumed in advance that these substances are not present in the DW - it must be checked.

In the revised version of the manuscript we have added the information’s about DW composition.

“Concentration of hydrogen peroxide, nitrites as well as nitrates in DW (i.e. without the plasma activation) was under the detection limit.”

  1. The chlorophyll concentration cannot be measured by N-tester. The N-tester measures the nitrogen content, not the chlorophyll content. The manufacturer's website says: ‘In theory N-Tester could be used on any plant to measure the chlorophyll content of the leaf and as such could be a useful management tool, however, no other crop calibrations exist so these would need to be developed locally by the user and any recommendations would need to be confirmed by locally conducted trials.’ In scientific research, such equipment for determining the concentration of chlorophyll should not be used.

In available literature it is stated that…Relative indices for the chlorophyll contents of leaves were obtained with the N-Tester chlorophyll meter.

  • Koning, L. A., Veste, M., Freese, D., & Lebzien, S. (2015). Effects of nitrogen and phosphate fertilization on leaf nutrient content, photosynthesis, and growth of the novel bioenergy crop Fallopia sachalinensis cv. ‘Igniscum Candy‘. Journal of Applied Botany and Food QualityVol 88 (2015), p.22–28. https://doi.org/10.5073/JABFQ.2015.088.005
  • Netto, A.L., Campostrini, E., Goncalves de Oliverira, J., BressanSmith, R.E., 2005: Photosynthetic pigments, nitrogen, chlrophyll a fluorescence and SPAD-502 readings in coffee leaves. Sci. Horti. 104 (1), 199-209. https://www.sciencedirect.com/science/article/pii/S030442380400189X

  1. I don't think the device used (VOLTCRAFT LCR 4080) is capable of measuring the electrical conductivity of the roots. Firstly, it is not dedicated to this type of measurement. Secondly, in other works using this device is an estimated root system size (https://doi.org/10.11118/actaun201866040871).

Standardly, an LCR meter is used to measure the root electric capacitance. We have the LCR meter VOLTCRAFT LCR 4080 available.

Linear correlation coefficients between electric capacitance and other parameters of the root system size are presented in:

  • Chloupek, O.,1972: Relationship between electric capacitance and some other parameters of plant roots.  Plant.14, 227—230. https://bp.ueb.cas.cz/pdfs/bpl/1972/03/08.pdf
  • Cseresnyés, I., Vozáry, E. & Rajkai, K. Does electrical capacitance represent roots in the soil?. Acta Physiol Plant42, 71 (2020). https://doi.org/10.1007/s11738-020-03061-9 http://real.mtak.hu/107947/1/s11738-020-03061-9.pdf
  • Chloupek O, Dostál V, Středa T, Psota V, Dvořáčková O (2010) Drought tolerance of barley varieties in relation to their root system size. Plant Breed 129:630–636. https://doi.org/10.1111/j.1439-0523.2010.01801.x
  • Postic, F., Doussan, C. Benchmarking electrical methods for rapid estimation of root biomass. Plant Methods 12, 33 (2016). 7https://plantmethods.biomedcentral.com/articles/10.1186/s13007-016-0133-7

At what level of soil moisture were these measurements carried out? The soil water content strongly influences the measurement results. Moreover, the measured values relate to the rhizosphere rather than the roots themselves.

The article states (L 151 – 153): “This instrument was used for the measuring of the root capacitances at two growth stages of plants (T2 and T4) in the pots after the irrigation (under condition of 100% of the full water capacity)”.

  1. Statistical analysis: maybe two-way ANOVA should be performed for all parameters tested, as experiments were established with two types of water (DW or PAW) and three number of water applications as experimental factors. In the methods authors must specify what type of ANOVA was performed.

We thank this Reviewer for this comment. The type of ANOVA (two-way) was added.

  1. L173-174 – the authors wrote: ‘While the N-tester value was decreased by the increasing intensity of the foliar DW application.’ – it is not true, statistical analysis does not support this statement.

  1. L203-204 – ‘Its mean value measured after the PAW application in all evaluated terms was almost identical with its value obtained with the DW application.’ – this sentence is redundant.

We thank the Reviewer for this comment. The sentence is redundant and was deleted.

  1. L281-282 – the authors wrote: ‘The enhancement of the mean values of the root capacitance was observed after the application of both types of water’, but what could be the reason for this? maybe the water from the leaves was just dripping down to the soil substrate?

We have no explanation for this result yet. It will be included in further experiments.

  1. L289- ‘The weight of the dry matter of the aboveground maize mass…’ – please, edit this part of the sentence, you can write it more simply.

We thank this Reviewer for this comment. The sentence was corrected and added to the manuscript.

“The dry matter weight of the aboveground maize mass, …”

  1. The authors wrote: ‘Nevertheless, from the mean values it is evident that the weight of the dry matter of the aboveground mass was decreased by the increasing doses/intensity of application.’; ‘An opposite trend was observed by the DW application when the weight of the dry matter was increased.’; ‘…higher PAW dose/intensity of application (PAW2 and PAW3) could be the cause of the decreased weight of the aboveground plant mass even by its foliar application.’; L318 ‘… lower weight of the dry matter in the variant PAW3….’- statistical analysis does not confirm these claims.

We thank this Reviewer for this comment. The authors believe that:Although the weight of the dry matter at the variant PAW3 is not significantly lower comparing to the others (PAW1-2), it does not mean that this fact cannot be the reason for the significantly higher nitrogen content in the plant after the application.

  1. Figures and tables: The letters and numbers in the figures are poorly visible. Their quality should be improved. Moreover, in my opinion, when there are no statistical differences between the means, it is better not to mark the means with letters, it creates unnecessary confusion. It is better to explain in the description that the lack of lettering indicates no significant differences.

We thank this Reviewer for this comment. The quality of figures was improved. The letters at the means with no statistical difference were left out.

  1. It is not known from how many data the means were calculated. The value of ‘n’ (e.g. n=30, n=4) should be provided in descriptions of tables and figures.

The values of “n” were added.

  1. Conclusions should not be a summary of the results. It should be emphasized what is the novelty of this research and what practical guidelines can be forwarded. This part is definitely too long.

The Conclusion part was shortened.

“Most of studies dealing with the PAW utilization in the plant production have tested its influence during the stage of germination or sprouting whereas realization was provided especially by the seed dipping or soil watering. There are almost none available data about the effect of its foliar application on the plant growth during the vegetation period. Obtained results shown a promising potential of the PAW application in the plant nutrition strategy. Due to the PAW composition, its application has a high potential not only in the plant nutrition where the periodical application positively influences the chlorophyll content in the maize leaves and the root production, but it has also decontamination effects on harmful factors, especially fungal diseases. Further research is supposed to be focused on the verification of the foliar PAW application in the plant protection and alternatively, as a substitution of water conventionally used for nutrient provision by the foliar fertilization.”

I am sending a revised and corrected manuscript “ Effect of Plasma Activated Water Foliar Application on Selected Growth Parameters of Maize (Zea Mays L.) " where we accepted all comments of the Reviewer.

Round 2

Reviewer 1 Report

I have no additional comments and I suggest the publication in present form.

Author Response

Dear Reviewer.

Thank you for your careful review of our paper, and for the comments, corrections and suggestions.

Best regards

Petr Škarpa

Reviewer 2 Report

The manuscript is generally improved from the earlier submission. However, I noticed a few issues/omissions and some sections are still incorrect. If these areas are addressed I would recommend the paper for publication.

1) The research hypothesis should take the form of a statement (not a question or guess). The hypothesis should always explain what you expect to happen. Rejecting the null hypothesis and accepting the alternative hypothesis is the basis for building a good research study. Please, correct accordingly.

2) The wording ‘The dry matter weight of the aboveground maize mass,…’ should be changed to ‘The dry weight of the aboveground organs of maize,…’

3) The authors wrote: ‘Each treatment was performed in eight replications (pots)’. Why then only 4 measurements were made for the analysed parameters? Measurements of the biomass of only 4 plants are not reliable, as well as measurements of the chlorophyll fluorescence and N-tester values. Why not all available biological objects were used?

4) The confusing letters with means that are no statistical different were not deleted (e.g. fig. 3, tab. 6). But the authors wrote that they did .... authors must first correct, then write that it has been done. Then the descriptions have to be modified accordingly. You must do your best to correct the manuscript.

5) Conclusions should include deductions from the conducted own research. In this form, these are assumptions that have not been confirmed by experiments e.g. ‘decontamination effects on harmful factors, especially fungal diseases’ were not determined in this study; what means ‘the plant nutrition strategy’ in the context of this study? These conclusions do not provide sufficient information that flow from this study.

Author Response

Dear Reviewer,

Thank you for your very careful review of our paper, and for the comments, corrections and suggestions. Below are the specific answers to the comments:

1) The research hypothesis should take the form of a statement (not a question or guess). The hypothesis should always explain what you expect to happen. Rejecting the null hypothesis and accepting the alternative hypothesis is the basis for building a good research study. Please, correct accordingly.

We thank the Reviewer for this comment. The research hypothesis was added.

“Based on chemical composition of the plasma activated water it can be assumed that its foliar application would influence a plant growth of maize, increase chlorophyll content and related parameters of photosynthesis, increase the aboveground organs production of plants as well as subsequent root system capacitance, and influence content of nutrients in plants.”

2) The wording ‘The dry matter weight of the aboveground maize mass,…’ should be changed to ‘The dry weight of the aboveground organs of maize,…’

We thank the Reviewer for this comment. The sentence was corrected.

3) The authors wrote: ‘Each treatment was performed in eight replications (pots)’. Why then only 4 measurements were made for the analysed parameters? Measurements of the biomass of only 4 plants are not reliable, as well as measurements of the chlorophyll fluorescence and N-tester values. Why not all available biological objects were used?

We thank the Reviewer for this comment. The eight measurements (in each term) were made for all chlorophyll fluorescence parameters, root capacitance, weight of dry mass and content of nutrients. The N-tester values were determined as the mean of 60 measurements in each term. The number of measurements was corrected.

4) The confusing letters with means that are no statistical different were not deleted (e.g. fig. 3, tab. 6). But the authors wrote that they did .... authors must first correct, then write that it has been done. Then the descriptions have to be modified accordingly. You must do your best to correct the manuscript.

We thank the Reviewer for this comment. We correct figure (fig 3) and table and the letters at the means with no statistical difference were left out.

5) Conclusions should include deductions from the conducted own research. In this form, these are assumptions that have not been confirmed by experiments e.g. ‘decontamination effects on harmful factors, especially fungal diseases’ were not determined in this study; what means ‘the plant nutrition strategy’ in the context of this study? These conclusions do not provide sufficient information that flow from this study.

We thank the Reviewer for this comment. The conclusions were corrected.

“Most of studies dealing with the PAW utilization in the plant production have tested its influence during the stage of germination or sprouting whereas realization was provided especially by the seed dipping or soil watering. There are almost none available data about the effect of its foliar application on the plant growth during the vegetation period. Due to the PAW composition, its application has a high potential in the plant nutrition. The periodical PAW application positively influences the chlorophyll content in the maize leaves. Content of chlorophyll expressed by the N-tester value was decreased in time at the highest distilled water application by 34.6% and by 24.7% at the PAW application of the same doses. The foliar PAW application significantly increased the root electrical capacitance, contrary to the weight of the aboveground biomass and the contents of nutrients in its dry matter except nitrogen. The periodical PAW application increased its content in the dry matter of the aboveground organs by 13.3%. The fluorescence parameters, that evaluate the efficiency of the photosystem II, were not significantly influenced by the PAW application, too. Further research is supposed to be focused on the verification of the foliar PAW application in the plant protection and alternatively, as a substitution of water conventionally used for nutrient provision by the foliar fertilization.”

Round 3

Reviewer 2 Report

The revised version of the MS looks appropreate. The authors have addressed all of my concerns.